# Yoga and Cardiovascular Health Trial (YACHT): a UK-based randomised mechanistic study of a yoga intervention plus usual care versus usual care alone following an acute coronary event

Therese Tillin ![ORCID],[1] Claire Tuson,[2] Barbara Sowa,[3] Kaushik Chattopadhyay,[4] Naveed Sattar,[5] Paul Welsh,[5] Ian Roberts,[6] Shah Ebrahim,[7] Sanjay Kinra,[7] A Hughes,[1] Nishi Chaturvedi[1]

For numbered affiliations see end of article.

**Correspondence to**
Dr Therese Tillin;
t.tillin@ucl.ac.uk

## ABSTRACT

**Objective** To determine the effects of yoga practice on subclinical cardiovascular measures, risk factors and neuro-endocrine pathways in patients undergoing cardiac rehabilitation (CR) following acute coronary events.

**Design** 3-month, two-arm (yoga +usual care vs usual care alone) parallel randomised mechanistic study.

**Setting** One general hospital and two primary care CR centres in London. Assessments were conducted at Imperial College London.

**Participants** 80 participants, aged 35–80 years (68% men, 60% South Asian) referred to CR programmes 2012–2014.

**Intervention** A certified yoga teacher conducted yoga classes which included exercises in stretching, breathing, healing imagery and deep relaxation. It was pre-specified that at least 18 yoga classes were attended for inclusion in analysis. Participants and partners in both groups were invited to attend weekly a 6- to 12-week local standard UK National Health Service CR programme.

**Main outcome measures** (i) Estimated left ventricular filling pressure (E/e′), (ii) distance walked, fatigue and breathlessness in a 6 min walk test, (iii) blood pressure, heart rate and estimated peak $VO_2$ following a 3 min step-test. Effects on the hypothalamus–pituitary–adrenal axis, autonomic function, body fat, blood lipids and glucose, stress and general health were also explored.

**Results** 25 participants in the yoga + usual care group and 35 participants in the usual care group completed the study. Following the 3-month intervention period, E/e′ was not improved by yoga (E/e′: between-group difference: yoga minus usual care:−0.40 (−1.38, 0.58). Exercise testing and secondary outcomes also showed no benefits of yoga.

**Conclusions** In this small UK-based randomised mechanistic study, with 60 completing participants (of whom 25 were in the yoga + usual care group), we found no discernible improvement associated with the addition of a structured 3-month yoga intervention to usual CR care in key cardiovascular and neuroendocrine measures shown to be responsive to yoga in previous mechanistic studies.

**Trial registration number** NCT01597960; Pre-results.

### Strengths and limitations of this study

► Comprehensive clinical and subclinical cardiovascular measures before and after a yoga intervention (plus usual cardiac rehabilitation (CR)) versus usual CR.
► Real world setting—older people following an acute coronary event.
► High level of dropout, particularly in the yoga plus usual CR arm.
► We can only assess the potential of yoga in addition to usual CR following an acute coronary event.
► Outcomes limited to 3 monthspost-intervention

## INTRODUCTION

The practice of yoga originated in ancient India as a form of exercise which includes breath control, the adoption of bodily postures and meditation which aim to increase strength and flexibility and to aid physical and mental well-being.[1] Yoga has been reported to reduce stress and depression and is thought to improve biological cardiovascular risk factors.[2–4] However, despite claims of benefits, the effects of yoga on cardiovascular outcomes remain unclear. Previous systematic reviews[5–12] confirm that investigations of the health benefits of yoga and underlying mechanisms have often been hampered by poor study design, including small sample sizes, inadequate adjustment for confounders, lack of randomisation, unsatisfactory masking of outcomes to assessors and publication bias. Also, many studies have been conducted in healthy young participants and it is not certain that these findings are generalisable to older adults with established disease.

Cardiac rehabilitation (CR) has been shown to improve cardiovascular mortality and hospital re-admissions in patients with coronary heart disease (CHD).[13] However, for myocardial infarction (MI), coronary bypass grafts and percutaneous coronary intervention (PCI) patients uptake to CR across in UK was only ~45% in 2012–2013 with low representation of ethnic minority people.[14] Yoga could therefore be a useful adjunct to CR.

In this UK-based randomised study (Yoga and Cardiovascular Health Trial (YACHT)), we hypothesised that yoga would be associated primarily with improvements in cardiovascular function and exercise capacity both chronically and acutely in people eligible for CR. The chronic study compared cardiovascular measures at 3 months between two groups randomised either to usual care (including CR) or to usual care plus a programme of yoga classes. For the chronic study, where the emphasis was on rehabilitation following a coronary event, we focused on the ratio between early mitral inflow and mitral annular early diastolic velocity (E/e′) as the primary cardiac measure. E/e′ provides an estimate of left ventricular (LV) filling pressure,[15] an aspect of LV diastolic function that predicts survival after MI.[16] We also performed a 6 min walk test (6MWT) as a measure of exercise tolerance and a 3 min step-test as a measure of cardiopulmonary fitness. These measures were chosen as they are reproducible and safe tests which are improved by CR,[17–20] and predict outcomes in people with CHD.[16 21–23] Measures chosen for the acute study (before and after the first session of yoga) included blood pressure (BP) and heart rate before and after exercise as indicators of cardiovascular and autonomic function which are associated with cardiovascular outcomes.[21 23]

In addition to these primary outcome measures, we studied a range of other cardiovascular risk factors and measures which might be expected to improve following CR and provide mechanistic insight into any beneficial effect of yoga; these included markers of the hypothalamic–pituitary axis, measures of autonomic function, measures of cardiac structure and function, brachial and central resting and 24-hour ambulatory BP, markers of atherosclerosis, blood glucose and lipids and self-reported health, lifestyle factors and perceived stress levels.

## METHODS
### Study population
Inclusion criteria included referral to CR programmes in north-west London following an acute coronary syndrome (MI, PCI, coronary artery bypass grafting). Pre-specified inclusion criteria were age between 35 and 80 years, male or female, without comorbid disease or mobility limitations that would preclude participation in CR and our investigations, and, given the north-west London area of recruitment, able to understand English or Punjabi. Ethnicity was self-defined, and verified by country of birth of all four grandparents. In all, 80 participants were recruited following discharge from

hospital and randomised in equal numbers to the yoga intervention plus their standard CR programme, or to standard CR programme (usual care) alone. Randomisation was performed by an independent researcher using a standard computerised algorithm (customised Java web application (srub)) and stratified by ethnicity (South Asian and non-South Asian), gender, 5-year age group and rehabilitation centre. The generated sequence was displayed only to the user at the time of assignment to the yoga intervention or usual care. 75% of participants were recruited from referrals to CR programmes at Ealing Hospital in west London, with the remainder recruited from two primary care CR programmes in north-west London (Harrow and Brent (Flexi-Heart Plan)). Recruitment of the planned 80 participants took place between October 2012 and April 2014, with the final participant seen for 3-month follow-up measures in July 2014.

Eligibility criteria were broadened in January 2013 and April 2013, respectively, with ethical approval, to include patients who had undergone coronary artery bypass grafting or who had received medical management only for their acute coronary event. The initial study plans were to recruit only patients referred to a CR programme post-angioplasty as treatment for an acute coronary syndrome. With cardiologist advice, it was felt that the earlier decision to exclude these patients based on safety grounds was unnecessary given the gentle and tailored nature of the exercises.

### Patient and public involvement
Patients and public were not involved in the study design, conduct, results, evaluation or dissemination.

Informed written consent was obtained from all participants.

### Yoga intervention
The yoga intervention was delivered on a twice-weekly group session basis for 12 weeks alongside the usual care, 6- to 12-week CR programme. There were 24 yoga classes in total. Participants' partners were invited to take part in each session as a method of improving adherence. The yoga session was designed and conducted by a teacher certified in yoga and CR, and included gentle exercises in deep relaxation, stretching, breathing, healing imagery and a healthy diet. A prescription of exercises with an accompanying DVD was provided to be performed regularly at home. Each session lasted approximately 75 min, divided into three equal parts: breathing exercises, yogic poses and meditations, education and discussion (details in online supplemental material: YACHT study package_v1.2.pdf). Individuals randomised to the yoga arm had their standard CR care delivered at a separate time to those randomised to usual CR (although delivered by the same teams), to reduce risks of contamination. Because the study was also designed to examine mechanisms underlying any beneficial effects of yoga,[24] there was a pre-specified requirement for participants in the yoga +usual care group to complete at least 18 yoga classes.

## Usual care

Usual care is described in the online supplemental material (YACHT study package_v1.2.pdf) and was similar in all centres in accordance with the UK's National Institute for Health and Care Excellence guidelines (https://www.nice.org.uk/guidance/cg48, accessed 25/8/2017) and British Association for Cardiac Prevention and Rehabilitation standards[25] with core components of lifestyle (physical activity, exercise, diet and weight management, smoking cessation), education, risk factor management, psychosocial, cardio-protective drug therapy and long-term management strategies. Patients and their partners were invited to attend once-weekly for a 6- to 12-week programme tailored to individual needs and including (1) ongoing risk factor monitoring/advice/support, (2) exercise sessions in a gym, led by cardiac physiologist or a home-based exercise programme, (3) health education lectures (led by CR sister, pharmacist, dietitian, clinical psychologist, cardiac physiologist), (4) relaxation sessions, (5) guidance and supervised use of the 'Edinburgh Heart Manual' (http://www.theheartmanual.com/Pages/default.aspx, accessed 26/9/2017).

## Outcome measures

### Chronic study: all measures performed pre-intervention and 3-month post-intervention
*Primary outcome measures*

#### Echocardiography

Transthoracic two-dimensional and Doppler echocardiography were performed as previously described.[26] Transmitral flow velocity during the early filling phase (E) was acquired by pulsed Doppler and averaged from three consecutive cycles. Tissue Doppler Imaging was performed on the lateral and septal LV wall. Peak velocities during early diastole (e′) were averaged from three consecutive representative cycles. The e′ wave velocities measured from the lateral and septal walls were averaged. The primary cardiac outcome was the ratio of early filling and early myocardial velocity (E/e′), a non-invasive estimate of LV filling pressure.[15]

#### Exercise capacity and physical fitness

Exercise capacity was measured by a 6MWT conducted along a 30 m straight path in an outdoor covered area marked clearly with the beginning and end of each lap. Participants wore appropriate shoes and loose-fitting clothing and rested in a chair for 10 min before the start of the test. Participants were asked to walk briskly as far as possible for a timed 6 min. Fatigue and dyspnoea before and after the walk test were assessed using the Borg scale.[27]

Physical fitness was measured using a Tecumseh step-test[28] Participants were asked to step repeatedly on and off a step measuring 60×30×17.5 cm (length, width and height) for 3 min in time with a metronome set to 92 beats per minute (bpm). This corresponds to a rate of energy expenditure approximately five times the basal metabolic rate.[29] Heart rate was measured on the right

arm immediately afterwards and then again in the seated position after 3 min recovery using an Omron 705CP device. Estimated peak oxygen consumption (peak $VO_2$) was calculated based on achieved heart rate in the immediate post-exercise period as described previously.[28]

### Chronic study: secondary outcome measures

Measures of cardiac structure and function were obtained as described under primary outcomes above and included LV mass index, relative wall thickness, left atrial diameter, ejection fraction, mitral E/A ratio, s′ (peak velocity during systole) and e′ (peak velocity during early diastole).

Resting seated brachial and central BP was measured after 5 min seated rest using a Pulsecor BP +device (Uscom Ltd, Sydney, Australia)[30] starting with the left arm and then repeated on the right arm. The average of the final two of three BP readings for the right arm were used, unless the average SBP was more than 10 mm Hg greater than the average in the left arm, in which case the left arm average readings were used as the measure of clinic BP. BP (standing) before, immediately after and after 3 min recovery following the step-test was measured using an Omron 705CP device on the right arm.

A Vicorder oscillometric device (SMT Medical Germany/Skidmore Medical UK)[31] was used to measure carotid-femoral pulse wave velocity (PWV).

Ambulatory BP monitoring was conducted using the oscillometric Mobilograph device (NuMed Healthcare, UK)[32] with an appropriately sized cuff worn on the non-dominant arm to record central BP and heart rate for a 24-hour period; measurements were taken half-hourly between 07:00 and 21:00 hours and hourly during the night. Ambulatory BP and heart rate analyses included the daytime period from 09:00 to 21:00 hours and the night-time period from 01:00 to 06:00 excluding the waking and bedtime periods of the day as these periods represent times during which bed rest is inconsistent and, therefore, cannot be categorised reliably.[33]

Heart rate variability (HRV) and baroreceptor sensitivity (BRS) were measured according to a published protocol.[34] Briefly, these were measured in the recumbent position for a 10 min period. Beat-to-beat arterial BP was recorded non-invasively using a Finometer (FMS Amsterdam, Netherlands), and the ECG was monitored using a three-lead ECG. Signals were post-processed as described in detail previously.[34] For HRV, we calculated the mean R-R interval, and mean spectral powers in the low-frequency (0.04–0.15 Hz) and high-frequency (0.15–0.4 Hz) bands for the R-R intervals. Frequency domain BRS was calculated as the alpha index given by the square root of the ratio between averaged powers of R-R and systolic BP for each frequency.

Fasting bloods were analysed for glucose and lipids at baseline and 3-month follow-up. The HPA axis was assessed by salivary cortisol sampled at five points during the day pre-intervention and at 3-month follow-up as described for the acute study below. Salivary amylase, as a

marker of sympathetic activity, was measured at five time points during the day, as described for cortisol.

The full extra-cranial carotid artery was examined for the presence of plaque using an iE33 ultrasound machine (Philips) equipped with a linear-array transducer (L11_3) with concurrent recording of three-lead ECG over three–five cardiac cycles. Carotid intima-media thickness (IMT) was measured in the distal 1 cm of the left common carotid artery from three longitudinal planes (anterior, lateral and posterior) in a region free of plaque with a clearly identified double-line pattern. Plaque was defined according to the Mannheim consensus as a focal structure encroaching into the arterial lumen by at least 0.5 mm or 50% of the surrounding IMT value, or a region of IMT having a thickness >1.5 mm. Analyses were performed using a validated semi-automated programme (AMS-II).

The GeneActiv wrist-worn waterproof accelerometry device was fitted at the end of the pre-intervention and 3-month follow-up visits and worn for 3 days after each visit. Analysis of the data was performed using a validated algorithm at the University of Newcastle[35] to provide average body acceleration (metric milli g where g is gravity) on days with more than 16 hours of valid readings.

Self-completion questionnaires were administered pre-interventions and at the 3month follow-up as follows:

The international physical activity questionnaire (IPAQ) long version was administered and analysed according to the IPAQ guidelines (http://www.ipaq.ki.se/scoring.pdf, accessed 25 August 2017)

A self-completion questionnaire included items regarding frequency of alcohol consumption, number of units consumed and changes in drinking habits. Similar questions were included regarding smoking habits. A food frequency questionnaire, previously used in the SABRE tri-ethnic cohort study[36] covered the previous 7 days.

Euroqol-5 dimension-3 Levels (EQ-5D-3L) (https://euroqol.org/) is a standardised instrument for use as a measure of health outcome. It provides a simple descriptive profile a visual analogue scale to indicate self-rated health and a health status score based on UK population norms (there is no set of scores based on Indian Asian populations).

The perceived stress 10-item self-completion scale[37] was completed together with questions regarding sleep quality, snoring and breathlessness at night.

### Acute study: Yoga+usual care group on day of first yoga session
*Primary outcome measures*
BP and heart rate at rest and following a 3 min step-test and estimated peak oxygen consumption (peak $VO_2$) measured immediately post-exercise.

Seated brachial BP was measured after 5 min rest using an Omron 705CP device on the right arm. BP, heart rate at rest and following the 3 min step-test were performed immediately before and after the first yoga session as described above for the chronic study; estimated peak $VO_2$ was also calculated.[28 29]

### Secondary outcome measures
Saliva samples for amylase and cortisol were collected by the participants at home using a Salivette (www.salimetrics.com) collection kit at five time points during the day pre-intervention (waking, waking plus 30 min, waking plus 90 min, waking plus 12 hours, bedtime). For the acute study, waking, waking plus 12 hours and bedtime samples were taken on the day of the first yoga session. The latter two sampling points therefore occurred after the first yoga session. Samples were analysed using using indirect ELISA kits (Salimetrics Europe, Suffolk, UK).

### Blinding of observers
Post-processing of echocardiograms, carotid ultrasound scans, accelerometry, ambulatory BP, HRV and BRS, blood and saliva analyses were all conducted by observers blinded to participants' identity and study group. Clinic BP, vascular measurements and anthropometric measurements were conducted by clinic staff, who may have been aware of study group allocation, given the nature of the interventions.

### Location where data were collected
Data were collected at the International Centre for Circulatory Health on the St Mary's campus of Imperial College London (UK).

### Statistical analyses
#### Sample size and power
The sample size estimate was based on primary outcome measures for the chronic effects of yoga, that is, E/e′ echocardiography, 6MWT and Tecumseh 3 min step-test. Previous studies have reported at least half a SD benefit associated with yoga on both diastolic function and exercise testing[38 39] corresponding to a 1.1 improvement in E/e′,[38 39] and a study in people with preserved ejection fraction heart failure reported more than double this effect (−3.2) following a 3-month exercise programme.[20] For exercise testing (the 6MWT), a distance of 40 m (equivalent to 0.5 SD benefit) was considered a clinically significant improvement in distance walked.[40] This improvement was exceeded in a study of CR, where the distance walked increased by 62 m.[41] In both cases, these minimum important differences corresponded to approximately 0.5 SD . The sample size was estimated to detect effects of this magnitude for the three primary outcomes. Statistical analyses were planned to use regression modelling to adjust final measures for baseline differences, thus improving the precision of estimates of treatment effect, and shrinking the sample size requirement.[42] The intraclass correlation coefficient (ICC) of the primary outcomes was ≥0.85 based on our own data (n=10) and other observers'.[19] Using a conservative estimate of ICC=0.70, and allowing for multivariable analysis, 33 completers were required in each arm of the study to detect a 0.5 SD difference between groups (80% power and 5% significance). Thus, 40 people were recruited to each arm to allow for dropouts.

**Table 1** Pre-intervention characteristics by randomisation group (unadjusted)

| N (%) or means (95% CI) unless otherwise stated | Yoga +usual care | Usual care | P value for between group difference |
|---|---|---|---|
| Pre-intervention | n=40 | n=40 | |
| Ethnicity: South Asian | 25 (63%) | 26 (65%) | 0.8 |
| Sex: male | 28 (70%) | 26 (67%) | 0.8 |
| Age: years | 57.4 (54.1–60.7), range (35–77) | 56.9 (53.8, 60.0), range (35–78) | 0.8 |
| Days since coronary event | 50 (43, 57) | 59 (53, 65) | 0.09 |
| Diabetes* (self report of physician diagnosis/anti-diabetic medication) | 15 (38%) | 14 (35%) | 0.8 |
| Hypertension*(self report of physician diagnosis) | 29/37 (78%) | 25/37 (68%) | 0.3 |
| Heart failure* (self report of physician diagnosis) | 7/29 (19%) | 7/29 (19%) | 1.0 |
| Antihypertensive medications* | 39 (98%) | 36 (90%) | 0.17 |
| Number of antihypertensive medications, median (95% CI)* | 3 (3,3) | 3 (3,3) | 0.8 |
| Beta-blockers* | 33 (83%) | 32 (80%) | 0.8 |
| Statins* | 36 (90%) | 36 (90%) | 1.0 |
| Current smoker/ex/never smoker, number* | 4/14/19 | 1/14/24 | 0.3 |
| Alcohol: never/ever drinkers, number* | n=36 Never drinkers: 13 Ever drinkers: 23 | n=35 Never drinkers: 10 Ever drinkers: 25 | 0.5 |
| Units/week (ever drinkers)*, median (95% CI) | 2.5 (0, 10) | 4 (1, 7) | 0.9 |
| Currently employed* | n=35 15 (43%) | n=32 15 (47%) | 0.7 |

*Self-reported, n=number of responses to questionnaire item if incomplete.

## Statistical methods

Chronic study: summary descriptions of continuous pre-intervention characteristics are shown as means (95% CI) for normally distributed data or as medians (95% CI of the median (CIM)) for non-normally distributed variables or as number (%) for categorical variables. Pre-intervention characteristics are shown for the whole study group (table 1) and for those who did and did not complete the study. (online supplementary table S1). Outcome analysis is restricted to those who attended the 3-month visit, and for the yoga group, additionally restricted to those who attended 18 out of the 24 yoga sessions, per protocol. A sensitivity analysis added four participants who did not complete the requisite number of yoga classes but who attended the 3-month study follow-up visit.

For the 3 min step-test which was conducted in three stages pre- and post-intervention, repeated measures analysis of variance (ANOVA) models were used to determine differences by intervention arm and timing (pre-intervention and 3-month follow-up for the chronic study) and for the acute study (pre-first and post-first yoga session). Repeated measures ANOVA models were also used for salivary amylase (log transformed) and cortisol

measured 5 times on 3 days (yoga +usual care group) or 2 days (usual care group).

The remaining measures were analysed using robust regression models,[43] which are relatively efficient in the presence of outlier-prone error distributions. Three-month follow-up values were adjusted for the pre-intervention value of each Normally distributed measure, to provide adjusted mean (95% CI) values to allow comparison with pre-intervention observations. Where data were not normally distributed pre-intervention, median regression provided comparable 3-month (median (95% CIM)) follow-up values adjusted for the pre-intervention value. We show between group differences (95% CI) and p values for all outcome measures. Sensitivity analyses for primary outcomes included adjustment for informative baseline covariates (age, sex, diabetes, body mass index plus height for the 6MWT). Between-group and within-group differences in categorical secondary outcome measures were tabulated and tested using the $\chi^2$ test.

For HRV and BRS, we conducted sensitivity analyses that excluded the few participants who were not receiving beta-blocker medication.

Statistical analyses were performed using STATA V.15 software.

## RESULTS

In all, 80 participants were recruited and randomly assigned in equal numbers to the yoga plus usual care and usual care groups. Pre-intervention, average age was 57.1 (95% CI: 54.9 to 59.4), 68% were men and 64% were of South Asian origin. Diabetes was present in 36%. The majority were receiving statins (90%) and/or anti-hypertensive medication (95%) (table 1). Consistent with current practice in the UK, 91% had received PCIs.

In all, 35 participants in the usual care arm (63% South Asian) and 25 participants in the yoga arm (59% South Asian) completed the study. Greater loss to follow-up occurred in the yoga group, mostly due to unwillingness to continue with yoga classes—participants frequently citing ill health as a reason, although one participant withdrew from the study because of return to work (online supplementary figure S1). Characteristics of those who completed the study and those who dropped out were similar pre-intervention (online supplementary table S1). In addition to overall study dropout, several participants declined or were unable to undergo exercise testing either pre-or post-intervention, mostly due to mobility problems or elevated BP (reasons are listed under table 2).

No adverse events were reported. There was minimal change in the number and type of medications prescribed over the 3-month course of the study (table 1 and online supplementary table S1).

### Chronic study
#### Primary outcomes
##### LV diastolic function

At 3-month follow-up, E/e′ improved in both groups, but there was no evidence of yoga-related additional benefit in diastolic function (E/e′: between-group difference: yoga minus usual care:−0.40 (−1.38 to 0.58) (adjusted for pre-intervention values) p=0.4 (table 2)

##### 6 Min walk test

The total distance walked increased in both groups at 3-month follow-up, but there was no evidence of yoga-related additional benefit (between group difference yoga minus usual care: −7 (−39, 26) m, p=0.7; table 2). Distance walked per minute also increased post-intervention to a similar level in both groups and there was no additional advantage related to yoga in the total number of minutes walked or in levels of fatigue and breathlessness (table 2)

##### 3 Min step-test

The results of the 3 min step-test at 3-month follow-up suggested some moderate improvements in immediate post-exercise BP, heart rate and peak $VO_2$ in both groups at follow-up, but there was no evidence of additional benefit associated with yoga (table 2).

#### Secondary outcomes
##### Other vascular measures

There was no evidence of yoga-related additional benefits for measures of clinic and ambulatory measures of brachial and central SBP at follow-up. Both groups showed improvements in resting brachial DBP and in resting central SBP. PWV was similar in the two groups at follow-up (table 3).

##### Carotid IMT

There was no evidence of additional yoga-related benefit on carotid IMT levels at 3 months (table 3).

##### HPA axis

Salivary cortisol, as a marker of the HPA, decreased throughout the day in both groups pre-intervention and at 3-month follow-up. There was no evidence of additional yoga-related benefit compared with usual care alone (table 3)

##### Autonomic function

There was no evidence of additional yoga-related benefit compared with usual care alone on markers of HRV, BRS at 3-month follow-up and salivary amylase (table 4).

##### Metabolic measures

There was no evidence of additional yoga-related benefit compared with usual care alone at 3-months follow-up in glucose, total cholesterol and low-density lipoprotein (LDL) cholesterol (table 3).

##### Anthropometrics

Both groups had slightly lower waist to hip ratios at follow-up than at baseline, but with no evidence of yoga-related additional benefit compared with usual care alone. (table 3).

#### Other measures

Accelerometry over 3 days showed that the usual care group modestly increased levels and the yoga group maintained levels of physical activity during the follow-up period. Self-reported physical activity (IPAQ) increased in both groups, with no evidence of additional benefit from yoga compared with usual care alone (table 3).

Similarly, the EQ-5D-3L measures of health status or self-rated health at follow-up did not show any convincing evidence of a treatment effect at follow-up, although there was a small increase in EQ-5D-3L health status based on UK population norms in people randomised to yoga compared with those receiving usual care. The EQ-5D-3L self-rated health thermometer improved to equal extents in both yoga and usual care groups over the 3-month period. The yoga group had lower stress scores than the usual care group both pre-intervention and at follow-up and there was no convincing evidence of change in stress score in either treatment group (table 3).

There were very few current smokers at baseline or follow-up and there were no between-group differences or within-group changes. There were no between-group

**Table 2** Chronic study—primary outcomes: recruitment and 3-month follow-up (includes only those who attended study clinics at both time points and attended at least 18 classes if in the yoga group: n=35 in usual care group and n=25 in yoga group, unless otherwise stated)

| | Pre-intervention Means (95% CI) | | 3-month follow-up Means (95% CI), adjusted for pre-intervention levels | | Between-group adjusted difference for pre-intervention levels (95% CI) | P value for between group difference, adjusted for pre-intervention levels |
|---|---|---|---|---|---|---|
| | Yoga +usual care | Usual care | Yoga +usual care | Usual care | Yoga minus usual care | |
| **Diastolic function** | n=25 | n=33* | n=25 | n=33* | | |
| E/e'. Median (95% CI) | 9.74 (8.37 to 11.12) | 8.72 (7.76 to 9.68) | 8.81 (8.33 to 9.29) | 8.26 (7.79 to 8.74) | −0.40 (−1.38 to 0.58) | 0.4 |
| **6 min walk test** | n=19† | n=30‡ | n=19† | n=30‡ | | |
| Total distance, m | 462 (449 to 517) | 442 (402,482) | 488 (463 to 513) | 491 (471 to 512) | −7 (−39 to 26) | 0.7 |
| Total minutes walked | 6.0 (6.0 to 6.0) | 5.8 (5.4 to 6.0) | 5.9 (5.7 to 6.1) | 5.8 (5.6 to 6.0) | 0.1 (−0.3 to 0.5) | 0.5 |
| Distance, m/min | 77 (72 to 82) | 77 (71 to 82) | 81 (78 to 83) | 81 (78 to 83) | 0.8 (−4.6 to 6.2) | 0.8 |
| Fatigue (Borg scale: 0–10) | | | | | | |
| Pre-test | 0.2 (0 to 0.7) | 0.2 (0 to 0.5) | 0.02 (−0.03 to 0.07) | 0 (0 to 0.06) | 0.07 (−0.03 to 0.2) | 0.17 |
| Post-test | 0.6 (0 to 1.4) | 0.7 (0.03 to 1.4) | 0.08 (−0.06 to 0.20) | 0.08(−0.04 to 0.19) | −0.01 (−0.3 to 0.2) | 0.9 |
| Dyspnoea (Borg Scale: 0–10) | | | | | | |
| Pre-test | 0 (0 to 0) | 0.07 (0 to 0.23) | 0 (0 to 0.09) | 0.04 (0 to 0.10) | 0.04 (−0.1 to 0.07) | 0.5 |
| Post-test | 0.6 (0 to 1.2) | 1.0 (0.3 to 1.7) | 0.2 (−0.3 to 0.8) | 0.6 (0.2 to 1.0) | 0.2 (−0.9 to 0.5) | 0.5 |
| **Response to exercise: 3 min step-test** | n=18† | n=30‡ | n=18† | n=30‡ | | |
| Pre-step-test | | | | | | |
| Brachial SBP, mm Hg | 140 (134 to 146) | 135 (130 to 140) | 138 (133 to 140) | 131 (126 to 136) | 6 (−5 to 18) | 0.3 |
| Brachial DBP, mm Hg | 83 (80 to 85) | 80 (78 to 83) | 82 (79 to 84) | 79 (77 to 82) | 2 (−4 to 9) | 0.4 |
| Heart rate, bpm | 60 (56 to 64) | 62 (59 to 66) | 58 (54 to 61) | 60 (57 to 63) | −2 (−8, 5) | 0.6 |
| Immediately post-step-test | | | | | | |

**Table 2** Continued

| | Pre-intervention Means (95% CI) | | 3-month follow-up Means (95% CI), adjusted for pre-intervention levels | | Between-group difference adjusted for pre-intervention levels (95% CI) | P value for between group difference, adjusted for pre-intervention levels |
|---|---|---|---|---|---|---|
| | Yoga +usual care | Usual care | Yoga +usual care | Usual care | Yoga minus usual care | |
| Brachial SBP, mm Hg | 161 (155 to 167) | 152 (147 to 157) | 156 (151 to 162) | 149 (145 to 155) | 2 (−10 to 14) | 0.7 |
| Brachial DBP, mm Hg | 85 (82 to 87) | 80 (78 to 83) | 77 (74 to 80) | 77 (74 to 79) | −1 (−8 to 5) | 0.7 |
| Heart rate, bpm | 89 (85 to 93) | 89(86 to 93) | 81 (77 to 85) | 85 (82 to 88) | −4 (−11 to 3) | 0.2 |
| Peak VO$_2$ mL/min/ kg | n=14 35.7 (31.5 to 40.0) | n=27 38.2 (34.3 to 42.1) | n=14 41.3 (38.0 to 44.6) | n=27 42.6 (39.3 to 45.8) | 0.4 (−5.9 to 6.6) | 0.9 |
| 3 min post-step-test | | | | | | |
| Brachial SBP, mm Hg | 143 (137 to 149) | 134 (129 to 139) | 144 (138 to 150) | 136 (131 to 140) | 4 (−8 to 15) | 0.6 |
| Brachial DBP, mm Hg | 83 (80 to 86) | 80 (78 to 83) | 80 (77 to 83) | 79 (77 to 82) | −1(−3 to 9) | 0.8 |
| Heart rate, bpm | 66 (62 to 70) | 65 (61 to 68) | 61 (57 to 65) | 60 (58 to 64) | 0 (−7 to 7) | 1.0 |

*Two missing cases due to error readings.
†Missing cases: test not performed due to poor mobility (2) elevated blood pressure(>180/100 mm Hg) (4) refused step-test (1).
‡Missing cases: test not performed due to poor mobility (3), recent myocardial infarction (1), refused (1).
bpm, beats per minute; DBP, diastolic blood pressure; SBP, systolic blood pressure.

**Table 3** Chronic study—secondary outcomes: pre-intervention and 3-month follow-up (includes only those who attended study clinics at both time points and attended at least 18 classes if in the yoga group; n=35 in usual care group and n=25 in yoga group, unless otherwise stated)

| | Pre-intervention Means (95% CI) unless stated otherwise | | 3-month follow-up, adjusted for pre-intervention levels. Means (95% CI) unless stated otherwise | | Between-group difference adjusted for pre-intervention levels (95% CI) | P value for between group difference, adjusted for pre-intervention levels |
|---|---|---|---|---|---|---|
| | Yoga +usual care | Usual care | Yoga +usual care | Usual care | Yoga minus usual care | |
| **Cardiac structure** | | | | | | |
| Left ventricular mass indexed to height $^{2.7}$, g/m $^{2.7}$, median (95% CI) | 40.1 (36.6 to 51.2) | 38.7 (35.6 to 43.5) | 39.4 (36.2 to 42.5) | 39.3 (36.1 to 42.5) | 3.4 (−3.3 to 10.1) | 0.3 |
| Relative wall thickness, median (95% CI) | 0.45 (0.38 to 0.48) | 0.42 (0.40 to 0.47) | 0.41 (0.39 to 0.44) | 0.42 (0.39 to 0.44) | −0.01 (−0.07 to 0.04) | 0.6 |
| Left atrial diameter indexed to height, cm/m | 2.2 (2.2 to 2.3) | 2.3 (2.2 to 2.4) | 2.2 (2.2 to 2.3) | 2.2 (2.2 to 2.3) | −0.08 (−0.21 to 0.05) | 0.2 |
| **Cardiac function** | | | | | | |
| Ejection fraction, %, median (95% CI) | 54 (44 to 68) | 54 (44 to 64) | 54 (44 to 63) | 54 (45 to 63) | −0.5 (−19 to 18) | 1.0 |
| Mitral E:A ratio (of peak velocity in early diastole to peak velocity in late diastole, median (95% CI) | 1.02 (0.97 to 1.12) | 1.16 (0.95 to 1.29) | 1.09 (1.01 to 1.18) | 1.14 (1.07 to 1.23) | −0.01 (−0.20 to 0.19) | 1.0 |
| s', peak velocity during systole, cm/s, median (95% CI) | 7.11 (6.49 to 8.25) | 7.31 (6.35 to 7.64) | 7.25 (6.90 to 7.59) | 7.16 (6.82 to 7.51) | 0.10 (−0.56 to 0.75) | 0.8 |
| e', peak velocity during early diastole, cm/s, median (95% CI) | 7.56 (6.77 to 9.35) | 8.57 (7.58 to 9.44) | 8.20 (7.73 to 8.67) | 8.45 (7.98 to 8.92) | 0.14 (−0.72 to 0.99) | 0.8 |
| **Resting blood pressure and heart rate** | | | | | | |
| Heart rate, bpm | 60 (55 to 64) | 64 (60 to 68) | 59 (57 to 61) | 61 (60 to 63) | −0.4(−4 to 3) | 0.8 |
| Brachial SBP, mm Hg | 134 (125 to 143) | 126 (120 to 133) | 127 (124 to 130) | 122 (119 to 125) | −2(−8 to 4) | 0.5 |
| Brachial DBP, mm Hg | 77 (74 to 80) | 74 (72 to 76) | 75 (73 to 76) | 73 (71 to 74) | −0.8(−3 to 2) | 0.6 |
| Central SBP, mm Hg | 128 (119 to 137) | 120 (114 to 125) | 122 (119 to 125) | 117 (114 to 120) | −1(−8 to 6) | 0.8 |

Continued

**Table 3** Continued

| | Pre-intervention Means (95% CI) unless stated otherwise | | 3-month follow-up, adjusted for pre-intervention levels. Means (95% CI) unless stated otherwise | | Between-group difference adjusted for pre-intervention levels (95% CI) | P value for between group difference, adjusted for pre-intervention levels |
| --- | --- | --- | --- | --- | --- | --- |
| | Yoga +usual care | Usual care | Yoga +usual care | Usual care | Yoga minus usual care | |
| **24-hour ambulatory blood pressure** | n=20 | n=30 | n=20 | n=30 | | |
| Average day central SBP, mm Hg | 115 (108 to 122) | 113(109 to 116) | 113 (110 to 116) | 112 (109 to 114) | 2 (-4 to 8) | 0.5 |
| Average night central SBP, mm Hg | 104 (96 to 112) | 104 (98 to 110) | 104 (100 to 108) | 104 (100 to 108) | 1 (-7 to 10) | 0.8 |
| Average day heart rate, bpm | 65 (60 to 69) | 68(65 to 71) | 64 (63 to 66) | 67 (65 to 69) | 2 (-2 to 5) | 0.3 |
| Average night heart rate, bpm | 59 (54 to 63) | 64 (60 to 67) | 59 (57 to 61) | 63 (61 to 65) | 4 (-0.4 to 7) | 0.08 |
| Pulse wave velocity, m/s median (95% CI) | n=25 9.03 (8.14 to 9.67) | n=32 8.63 (8.27 to 9.10) | n=25 9.02 (8.47 to 9.59) | n=32 8.75 (8.28 to 9.22) | 0.5 (-0.7 to 1.7) | 0.3 |
| CIMT. Far wall, mm, maximum of means (without plaque and CIMT <1.5 mm) | n=18 0.754 (0.691 to 0.816) | n=28 0.746 (0.681 to 0.810) | n=18 0.764 (0.699 to 0.830) | n=28 0.761 (0.696 to 0.827) | -0.1 (-0.2 to 0.1) | 0.4 |
| **Bloods, fasting, medians (95% CIM)** | n=25 | n=31 | n=25 | n=31 | | |
| Triglycerides, mmol/L | 1.10 (0.94 to 1.40) | 1.03 (0.96 to 12.4) | 1.11 (1.01 to 1.20) | 1.10 (1.00 to 1.19) | -0.04 (-0.24 to 0.17) | 0.7 |
| HDL cholesterol, mmol/L | 0.91 (0.84 to 1.07) | 0.91 (0.79 to 1.01) | 1.02 (0.97 to 1.06) | 0.97 (0.92 to 1.02) | 0.05 (-0.04 to 0.14) | 0.3 |
| Total cholesterol, mmol/L | 3.00 (2.80 to 3.50) | 3.20 (2.90 to 3.60) | 3.30 (3.18 to 3.41) | 3.36 (3.25 to 3.47) | 0.07 (-0.29 to 0.15) | 0.6 |
| Cholesterol:HDL ratio | 3.33 (2.82 to 3.67) | 3.40 (3.23 to 3.78) | 3.31 (3.13 to 3.48) | 3.52 (3.34 to 3.69) | -0.06 (-0.38 to 0.26) | 0.7 |
| LDL cholesterol, mmol/L | 1.59 (1.35 to 1.77) | 1.60 (1.54 to 1.73) | 1.76 (1.66 to 1.87) | 1.81 (1.71 to 1.92) | 0.01 (-0.19 to 0.20) | 0.9 |
| Glucose, mmol/L | 5.40 (4.90 to 6.0) | 5.50 (4.90 to 6.20) | 5.78 (5.62 to 5.94) | 5.68 (5.52 to 5.84) | 0.05 (-0.23 to 0.33) | 0.7 |
| **Anthropometrics** | n=25 | n=35 | n=25 | n=35 | | |

**Table 3** Continued

| | Pre-intervention Means (95% CI) unless stated otherwise | | 3-month follow-up, adjusted for pre-intervention levels. Means (95% CI) unless stated otherwise | | Between-group difference adjusted for pre-intervention levels (95% CI) | P value for between group difference, adjusted for pre-intervention levels |
|---|---|---|---|---|---|---|
| | Yoga +usual care | Usual care | Yoga +usual care | Usual care | Yoga minus usual care | |
| Body mass index, kg/m², median (95% CI) | 27.6 (25.1 to 29.5) | 27.2 (25.3 to 29.6) | 27.6 (27.4 to 27.9) | 27.5 (27.3 to 27.7) | 0 (−0.5 to 0.5) | 1.0 |
| Waist:hip ratio | 0.99 (0.96 to 1.03) | 0.99 (0.96 to 1.02) | 0.98 (0.97 to 0.99) | 0.98 (0.97 to 0.98) | 0.01 (−0.01 to 0.02) | 0.3 |
| Fat mass per cent | 28 (25 to 32) | 30 (27 to 33) | 28 (27 to 28) | 30 (29 to 30) | 0.2 (−1 to 2) | 0.8 |
| **HPA axis** | | | | | | |
| Salivary cortisol nmol/L | n=23 | n=29 | n=23 | n=29 | | |
| Waking | 11.2 (8.5 to 13.8) | 12.6 (10.3 to 14.9) | 11.2 (8.7 to 13.7) | 12.5 (10.4 to 14.7) | −1.3 (−4.2 to 1.6) | 0.5 |
| Waking +30 min | 10.6 (8.0 to 13.3) | 16.0 (13.6 to 18.3) | 10.4 (8.0 to 12.9) | 12.9 (10.7 to 15.0) | −2.5 (−5.4 to 10.5) | 0.17 |
| Waking +1 hour 30 min | 5.2 (2.5 to 7.8) | 8.0 (5.6 to 10.4) | 8.0 (5.6 to 10.4) | 7.5 (5.3 to 9.7) | 0.5 (−2.4 to 3.4) | 0.8 |
| Waking +12 hours | 3.9 (1.2 to 6.7) | 3.2 (0.8 to 5.7) | 4.1 (1.7 to 6.6) | 2.5 (0.3 to 4.7) | 1.6 (−1.4 to 4.5) | 0.4 |
| Bedtime | 2.3 (0 to 5.0) | 3.1 (0.7 to 5.5) | 3.7 (1.3 to 6.2) | 2.2 (0.04 to 4.3) | 1.6 (−1.3 to 4.5) | 0.4 |
| **Exercise/physical activity** | | | | | | |
| **Average body acceleration over 3 days**, milli g (GeneActiv) | n=20 25.1 (20.4 to 29.8) | n=28 22.5 (19.5 to 25.5) | n=20 24.9 (22.7 to 27.1) | n=28 23.5 (21.3 to 25.7) | −4.0 (−8.3 to 0.23) | 0.064 |
| International Physical Activity Questionnaire self-report. Total metabolic equivalent(met) min/week. Median (95% CI) | n=25 693 (60 to 1386) | n=34 1409 (495 to 2310) | n=25 2273 (1434 to 3112) | n=34 2899 (2065 to 3734) | 179 (−1542 to 1900) | 0.8 |
| EQ-5D health status based on UK population norms (1=full health) median (95% CI) | n=14 0.77 (0.69 to 1.0) | n=21 0.80 (0.73 to.81) | n=14 0.83 (0.70 to 0.97) | n=21 0.80 (0.66 to 0.93) | 0 (−0.24 to 0.24) | 1.0 |

**Table 3** Continued

| | Pre-intervention Means (95% CI) unless stated otherwise | | 3-month follow-up, adjusted for pre-intervention levels. Means (95% CI) unless stated otherwise | | Between-group difference adjusted for pre-intervention levels (95% CI) | P value for between group difference, adjusted for pre-intervention levels |
|---|---|---|---|---|---|---|
| | Yoga +usual care | Usual care | Yoga +usual care | Usual care | Yoga minus usual care | |
| EQ-5D self-rated health thermometer (100=best possible, median (95% CI) | n=21 70(60 to 75) | n=27 70 (50 to 80) | n=25 73 (68 to 78) | n=27 73 (68 to 78) | 2.5 (-5.8 to 10.9) | 0.6 |
| **Perceived stress score** (possible range: 0–40, 13 is considered average, high stress groups: 20+) | n=25 14.9 (11.8 to 18.0) | n=34 18.2 (15.2 to 21.2) | n=25 14.7 (13.1 to 16.4) | n=34 17.2 (15.5 to 18.8) | −2.7 (-6.1 to 5.1) | 0.11 |

bpm, beats per minute; CIMT, carotid intima media thickness; DBP, diastolic blood pressure; HDL, high-density lipoprotein; HPA, hypothalamus–pituitary–adrenal; LDL, low-density lipoprotein; SBP, systolic blood pressure.

differences or significant within-group changes at follow-up in self-reported hours and quality of sleep, in alcohol consumption or in consumption of fresh fruit and vegetables (not shown).

### Sensitivity analyses

Sensitivity analyses of the primary outcomes which added those four participants who did not complete 18 yoga classes, but who did attend the 3-month follow-up clinic, did not alter findings. Likewise, exclusion of the few people who were not receiving beta-blocker medication did not alter the findings for HRV, BRS and salivary amylase. Additional adjustment of primary outcome measures for selected informative baseline covariates (age, sex, diabetes, body mass index (and height for the 6MWT)) did not alter conclusions, for example, adjusted between group difference in E/e′ was −0.18 (−1.28, 0.92) compared with −0.38 (−1.38, 0.58) when adjusted only for baseline E/e′. The 3 min step-test and 6MWT findings were little changed on adjustment for these baseline covariates

### Acute study (yoga arm only)

A 3 min step-test was performed before and after the first yoga session and BP was measured pre-exercise, immediately post-exercise and 3 min post-exercise. In all, 27 participants undertook this test, three refused, eight were unable to undertake exercise testing due to mobility problems and/or shortness of breath, one had unstable angina and in one case equipment failure resulted in loss of data. There was no convincing evidence of an acute effect of yoga on BP, heart rate or estimated peak $VO_2$ (table 5a). Salivary cortisol and amylase were similar at the waking +12 hours and bedtime periods after the first yoga session compared with pre-intervention levels (table 5b).

### DISCUSSION

We show no additional cardiovascular benefit of a 3-month yoga intervention over and above usual care including CR in a randomised trial in people who had experienced an acute coronary event. Specifically, there was no additional impact on our co-primary outcomes of E/e′ or exercise capacity, nor on a wide range of other secondary outcomes including measures of cardiac structure and function, brachial, central and ambulatory BP, BP and heart rate responses to exercise, estimated peak $VO_2$, carotid IMT, blood lipids and glucose, obesity measures including fat mass and body mass index, self-reported physical activity levels, distance walked in the 6MWT alcohol, smoking and dietary intake.

Of the cardiovascular risk factors studied to date, BP appears the most consistently beneficially affected by yoga,[8 10 44] with reports that reductions in BP are similar to those obtained by anti-hypertensive medication.[45] However, a community-based crossover study in India of non-pharmacological interventions showed that physical exercise (brisk walking for 50–60 min, 3–4 days a week

**Table 4** Chronic study: autonomic function: HRV, BRS and salivary amylase

| | Pre-intervention | | 3-month follow-up adjusted for pre-intervention level | | Between-group difference adjusted for pre-intervention levels (95% CI) | P value for between-group difference adjusted for pre-intervention levels |
|---|---|---|---|---|---|---|
| | Yoga +usual care | Usual care | Yoga +usual care | Usual care | Yoga minus usual care | |
| **HRV and BRS 10 min recording** medians (95% CIM) | n=24 | n=33 | n=24 | n=33 | n=33 | |
| Number of beats | 677 (605 to 877) | 709 (632 to 798) | 644 (589 to 698) | 638 (584 to 693) | −1 (−110 to 108) | 1.0 |
| Number of ectopics | 19 (8 to 71) | 22 (8 to 48) | 29 (6 to 53) | 26 (2 to 49) | −1 (−46 to 44) | 1.0 |
| Mean RR interval, millisecond(ms) | 1016(888 to 1128) | 977 (925 to 1073) | 1050 (1012 to 1089) | 1020 (982 to 1059) | 13 (−62 to 87) | 0.7 |
| SDNN, ms | 55.5 (39.7 to 78.6) | 53.4 (37.4 to 69.2) | 48.2 (36.7 to 59.6) | 47.2 (35.9 to 58.6) | 3.6 (−20.4 to 27.5) | 0.8 |
| RMSSD, ms | 38.5 (30.0 to 60.0) | 38.8 (30.2 to 57.8) | 44.4 (35.2 to 53.5) | 39.9 (30.8 to 48.9) | 4.4 (−13.8 to 22.8) | 0.6 |
| NN50 | 40 (18 to 57) | 49(24 to 67) | 49 (31 to 66) | 51 (34 to 68) | 17 (−20 to 54) | 0.4 |
| pNN50 | 0.08 (0.03 to 0.11)) | 0.08 (0.04 to 0.14) | 0.14 (0.10 to 0.17) | 0.11 (0.07 to 0.15) | 0.01 (−0.06, to 0.08) | 0.7 |
| Triangular index | 179 (149 to 231) | 177 (139 to 235) | 189 (163 to 216) | 187 (160 to 214) | 20 (−37 to 77) | 0.5 |
| Total RR interval power, ms$^2$ | 1514 (633 to 2339) | 995 (680 to 2097) | 1157 (653 to 1662) | 1171 (668 to 1673) | 401 (−591 to 1393) | 0.4 |
| LF RR interval power, ms$^2$ | 340 (119 to 613) | 284 (177 to 491) | 340 (143 to 537) | 340 (143 to 538) | 20 (−391 to 430) | 0.9 |
| HF RR interval power, ms$^2$ | 251 (77 to 759) | 235 (126 to 317) | 265 (113 to 417) | 265 (113 to 16) | 101 (−256 to 457) | 0.6 |
| LF/HF power ratio | 1.5 (0.8 to 2.1) | 1.4 (1.1 to 1.7) | 1.2 (0.9 to 1.4) | 1.2 (0.9 to 1.4) | 0.04 (−0.5 to 0.6) | 0.9 |
| LF RR interval power, normalised units(nu) | 0.23 (0.16 to 0.32) | 0.27 (0.25 to 0.30) | 0.28 (0.24 to 0.31) | 0.29 (0.26 to 0.32) | 0.03 (−0.03 to 0.09) | 0.3 |
| HF RR interval power, nu | 0.17 (0.12 to 0.33) | 0.19 (0.14 to 0.28) | 0.23 (0.17 to 0.29) | 0.23 (0.17 to 0.29) | −0.02 (−0.13 to 0.10) | 0.8 |
| LF alpha index, ms/mm Hg | 6.9 (4.6 to 12.9) | 9.1 (6.6 to 13.6) | 10.1 (7.5 to 12.7) | 10.4 (7.8 to 13.0) | 0.8 (−4.4 to 6.1) | 0.8 |
| HF alpha index, ms/mm Hg | 9.2 (4.6 to 25.6) | 13.9 (9.2 to 22.6) | 15.8 (7.7 to 23.9) | 16.4 (8.3 to 24.5) | 3.7 (−9.4 to 16.8) | 0.6 |
| BRS on sequence analysis, ms/mmHg | 9.9 (6.3 to 16.1) | 10.0 (7.0 to 13.0) | 10.6 (8.2 to 13.1) | 10.6 (8.2 to 13.0) | 1.5 (−2.6 to 5.6) | 0.5 |
| **Salivary amylase, log microunits/L,** means (95% CI)) | n=23 | n=32 | n=23 | n=32 | | |

**Table 4** Continued

| | Pre-intervention | | 3-month follow-up adjusted for pre-intervention level | | Between-group difference adjusted for pre-intervention levels (95% CI) | P value for between-group difference adjusted for pre-intervention levels |
|---|---|---|---|---|---|---|
| | Yoga +usual care | Usual care | Yoga +usual care | Usual care | Yoga minus usual care | |
| Waking | 4.19 (3.83 to 4.41) | 4.48 (4.23 to 4.73) | 4.39 (4.06 to 4.71) | 4.45 (4.17 to 4.74) | −0.05 (−0.68 to 0.58) | 0.9 |
| Waking +30 min | 4.03 (3.74 to 4.33) | 4.08 (3.83 to 4.33) | 4.03 (3.71 to 4.34) | 4.26 (3.98 to 4.54) | −0.23 (−0.84 to 0.39) | 0.5 |
| Waking +1 hours 30 min | 4.45 (4.16 to 4.74) | 4.88 (4.62 to 5.13) | 4.48 (4.16 to 4.79) | 4.47 (4.19 to 4.76) | −0.01 (−0.63 to 0.61) | 1.0 |
| Waking +12 hours | 4.61 (4.30 to 4.92) | 4.74 (4.48 to 4.99) | 4.29 (3.97 to 4.61) | 4.70 (4.42 to 4.98) | −0.42 (−1.0 to 0.21) | 0.19 |
| Bedtime | 4.42 (4.12 to 4.72) | 4.66 (4.41 to 4.92) | 4.56 (4.23 to 4.88) | 4.44 (4.16 to 4.73) | 0.11 (−0.52 to 0.73) | 0.7 |

BRS, baroreceptor sensitivity; HF alpha index, high-frequency alpha index, ms/mm Hg; HF RR interval power, Relative power of the high-frequency band (0.15–0.4 Hz), nu; HF RR interval power, absolute power of the high-frequency band (0.15–0.4 Hz, ms²; HRV, heart rate variability; LF alpha index, Low-frequency alpha index, ms/mm Hg; LF/HF power ratio, ratio of LF-to-HF power; LF RR interval power, relative power of the low-frequency band (0.04–0.15 Hz), normalised units (nu); LF RR interval power, absolute power of the low-frequency band (0.04–0.15 Hz), ms², NN50, number of successive N-N intervals that differ by more than 50 ms; pNN50, Proportion of successive RR intervals that differ by more than 50 ms; RMSSD, root mean square of successive R-R interval differences, ms; SDNN, SD of N-N intervals, ms.

**Table 5a** Acute study, primary outcomes: immediate post-exercise results following a 3 min step-test, before and after (same day) the first yoga class

| n=27 | Before first yoga session Mean (95% CI) | After first yoga session: Mean (95% CI) | Post–pre-first yoga session difference mean (95% CI) | P value for comparison between before and after first yoga session |
|---|---|---|---|---|
| **Resting seated** | | | | |
| Systolic blood pressure, mm Hg | 132 (128 to 136) | 133 (129 to 136) | 1 (–9 to 11) | 0.9 |
| Diastolic blood pressure, mm Hg | 78 (77 to 81) | 79 (77 to 81) | 1 (–5 to 7) | 0.7 |
| Heart rate, bpm | 69 (67 to 71) | 68 (66 to 70) | –1 (–8 to 6) | 0.9 |
| **Immediately post-step-test, standing** | | | | |
| Systolic blood pressure, mm Hg | 161 (157 to 164) | 156 (148 to 163) | –5 (–15 to 6) | 0.4 |
| Diastolic blood pressure, mm Hg | 80 (78 to 82) | 80 (78 to 82) | –1 (–7 to 5) | 0.8 |
| Heart rate, bpm | 92 (90 to 94) | 88 (86 to 90) | –3 (–10 to 4) | 0.4 |
| Peak VO$_2$, mL/min/kg (n=24) | 36.9 (33.4 to 40.4) | 38.9 (35.4 to 42.4) | 2.0 (–0.2 to 4.1) | 0.07 |
| **3 min post-step-test, seated** | | | | |
| Systolic blood pressure, mm Hg | 133 (129 to 136) | 135 (131 to 139) | 2 (–8 to 13) | 0.7 |
| Diastolic blood pressure, mm Hg | 78 (76 to 80) | 79 (77 to 82) | 1 (–5 to 8) | 0.6 |
| Heart rate, bpm | 72 (70 to 74) | 69 (67 to 71) | –3 (–10 to 4) | 0.4 |

*12 participants did not participate in the step-test both before and after the first yoga session due to: mobilityproblems±shortness of breath (n=3), refused (n=3), frailty (n=2), unstable angina (n=1), other (high blood pressure/dizziness/weakness on left side/restricted movement; n=3). Blood pressure readings were unavailable for one participant due to equipment failure.

**Table 5b** Acute study: secondary outcomes: salivary amylase and cortisol: pre-intervention and day of first yoga class

| | Pre-intervention Means (95% CI) | Day of first yoga session Means (95% CI) | Post–pre-first yoga session difference Means (95% CI) |
|---|---|---|---|
| | n=32* | n=32* | |
| **Cortisol, nmol/L** | | | |
| Waking | 12.7 (11.0 to 14.4) | 15.4 (12.8 to 17.9) (pre-yoga) | N/A |
| 12 hours after waking | 3.6 (1.8 to 5.4) | 4.4 (1.7 to 7.0) (post-yoga) | 0.5 (–3.5 to 4.5) |
| Bedtime | 3.4 (1.6 to 5.2) | 3.0 (0.3 to 5.7) (post-yoga) | –0.3 (–4.3 to 3.7) |
| **Amylase, log microunits/L** | | | |
| Waking | 4.40 (4.23 to 4.57) | 4.28 (4.02 to 4.54) (pre-yoga) | N/A |
| 12 hours after waking | 4.66 (4.47 to 4.84) | 4.58 (4.32 to 4.85) (post-yoga) | –0.08 (–0.54 to 0.38) |
| Bedtime | 4.60 (4.42 to 4.78) | 4.74 (4.47 to 5.01) (post-yoga) | 0.13 (–0.32 to 0.59) |

*Eight participants were unable to provide adequate saliva samples pre-recruitment and on the day of the first yoga session. N/A, not available.

for 8 weeks) was a more effective method of reducing BP, compared with yoga training or salt reduction, which both had similar smaller effects.[46] More recently, a community-based randomised controlled trial in Sweden found no evidence of reductions in resting BP due to a 3-month yoga intervention[47]—similar to our findings.

The acute effects of yoga on cardiovascular responses to exercise have not been well studied, although a study

of 33 female college students in the USA reported that reduction in salivary cortisol following 1-hour sessions of power yoga, stretch yoga or control (watching an educational movie for 1 hour) were similar between interventions.[48] We saw little change in cortisol levels in the either group at 3-month follow-up or in the yoga group although later in the day following the first yoga session. At 3-month follow-up, improvements in heart rate and

estimated peak VO$_2$ were seen in both groups compared with the pre-intervention levels but there was no evidence of benefit from yoga. Likewise, both groups improved in terms of distance walked in the 6MWT at 3 months of follow-up but addition of yoga to usual care had no effect compared with usual care alone. An USA-based study in heart failure patients reported a 0.5 SD improvement in exercise tolerance (+17% in nine patients with heart failure enrolled to an 8-week yoga programme and −7% in 10 patients enrolled to receive standard medical therapy alone[39]). The same study also showed greater improvements in quality of life in the yoga group (scores improved by 26% in the yoga group and by 3% in the standard medical therapy group[39]). Heart failure was relatively infrequent in our study participants (less than 20% reported previously diagnosed heart failure and the median ejection fraction was 54% which is only slightly below the reported lower limit of the normal reference ranges for male and female Europeans (55.8% in men and 57.3% in women)[49]); but whether this can account for differences between study findings must remain speculative.

Measures of HRV may be more sensitive to subtle changes than traditional tests of autonomic function, and improvements associated with yoga training or tai chi have been observed in systematic reviews for a number of parameters[50 51]; however, we found little evidence for any yoga-related benefits in these parameters over 3 months.

Yoga has been variously shown to reduce fasting glucose and glycated haemoglobin, insulin, total and LDL cholesterol, triglyceride and weight, even in those without diabetes,[52] although not all studies have shown a consistent benefit across these risk factors.[7 53 54] We found no evidence of yoga-related benefits in blood lipids, glucose or obesity measures. Statin use was high (89%+) in both our study groups which may limit the measurable effect of interventions on lipid levels.

A 3-month multimodality intervention including yoga improved stress management, stress, depression, hostility and quality of life in a large group of patients with CHD, although this study lacked a control arm.[55 56] While we showed little change in perceived stress and did not measure stress management, depression or hostility, there was a small improvement in the EQ-5D measure of health status based on UK population norms in the yoga group in our study although the 95% CI of the difference included the null.

Impacts of yoga on subclinical and clinical cardiovascular disease have been inconsistently studied, making it difficult to place in context our findings of no evidence of improvement in E/e′ due to yoga, although E/e′ appeared lower in both groups at follow-up compared with baseline. Although our participants had pre-intervention levels of E/e′ that could be considered 'normal', it is important to note that increased cardiovascular risk increases linearly across the entire range of E/e′ and there is evidence that exercise can improve diastolic function even in healthy individuals.[57] Yoga in comparison with walking has been reported to improve cardiac function in older hypertensive individuals in India[58] and, in a high-risk subgroup of older individuals in the USA a multimodality intervention including yoga reduced carotid IMT to a greater extent than usual care or dietary and exercise advice[53]; however, no effect of the intervention was seen in the whole group, and numbers in the high-risk subgroups were small. Although our study was not designed or powered for long-term follow-up of cardiac events, we note that a multimodality intervention including yoga, in a group of 35 participants, showed reduced numbers of cardiac events and progression of atherosclerosis to a greater extent in the lifestyle intervention arm compared with the control arm over 5 years of follow-up.[59] Similarly in 123 angiographically documented patients with moderate to severe coronary artery disease, the Mount Abu Open Heart Trial of a multimodality intervention including Rajyoga meditation showed a trend towards regression of coronary lesions and a reduction in coronary events in those most adherent to the programme compared with those least adherent.[60]

As noted earlier, there is a general difficulty in comparing studies due to the wide variation in study designs and populations. A recent systematic review of 306 randomised controlled trials of yoga found that 91% reached positive conclusions.[61] The authors confirmed difficulty in comparison of results across all trials, due to the common lack of a priori defined primary outcomes and appropriate group comparisons.

### Strengths and limitations

This is the first study to our knowledge to adopt a comprehensive approach to measuring cardiovascular clinical and subclinical outcomes in response to a yoga intervention. It is also unusual in studying outcomes in a real-world setting in an older group of people eligible for CR following an acute coronary event; however, the trial was not designed to establish whether yoga may have benefits in terms of cardiovascular events or angina, for this much larger studies would be required. Also for some outcomes, such as IMT, it is likely that 3 months is too short a time to observed substantial regression. Dropout in the yoga arm of the study exceeded that in the usual care arm (25 and 35 completed the study, respectively), possibly reflecting the dual burden of attending both yoga training and usual CR. Consequently, the study will not have achieved the planned statistical power, as we had estimated a requirement for 33 in each group to enable detection of a 0.5SD difference in the primary outcomes, although given the measured effect sizes and CIs, we believe if there are benefits of yoga on the measured outcomes, they are likely to be small. We did not adjust for multiple testing as we had identified a priori relevant primary outcomes for the trial, but in practice adjustment for multiple testing would not have altered our interpretation given the null findings. CR is standard care following an acute coronary event in the UK, thus ethical reasons prevented comparison of yoga-based CR

directly with usual CR alone—hence, this study cannot tell us about the potential of yoga as an alternative to traditional CR. Our findings also cannot be generalised to other conditions that may benefit from CR, such as heart failure, post-valve replacement, stable angina pectoris or symptomatic peripheral artery disease.[25] It should also be noted that our study was designed as a mechanistic study to complement a larger (around 4000 patients) Indian Council for Medical Research and Medical Research Council, UK-funded study of yoga as a primary method of CR in India, which may shed light on some of the issues discussed above.[24]

## CONCLUSION

In this small UK-based randomised mechanistic study with 60 completing participants (of whom 25 were in the yoga +usual care group), we found no discernible improvement associated with the addition of a structured 3-month yoga intervention to usual CR care in any key cardiovascular or neuroendocrine outcomes shown to be responsive to yoga in previous mechanistic studies.

We suggest that usual care CR programmes in the UK, which include exercise, and optimisation of medical therapy may leave little additional scope for added benefits from a further intervention such as yoga.

**Author affiliations**
[1]Department of Population Science and Experimental Medicine, Institute of Cardiovascular Science, University College London, London, UK
[2]York Hospitals NHS Trust, York, UK
[3]West London Mental Health NHS Trust, Southall, UK
[4]Division of Epidemiology and Public Health, School of Medicine, University of Nottingham, Nottingham, UK
[5]Institute of Cardiovascular and Medical Sciences, University of Glasgow, Glasgow, UK
[6]Department of Population Health, London School of Hygiene and Tropical Medicine, London, UK
[7]Department of Non-communicable Disease Epidemiology, London School of Hygiene and Tropical Medicine, London, UK

**Acknowledgements** We thank the cardiac rehabilitation teams at Ealing Hospital and the Flexi-Heart cardiac rehabilitation teams in Harrow and Brent for their help in identifying suitable participants; Daniel Key for building the study databases and managing the randomisation processes; Paula Carvelli, April McGowan for their help in recruitment and clinical measurements; Nadia de Couto Franscisco and Karikahan Manoharan for echocardiography measurements and analysis; Vincent van Hees (www.esciencecenter.nl) for performing analyses of GeneActiv physical activity data; and the participants for their valuable support.

**Contributors** SK, NC, AH, SE and IR were involved in the conception and design of the study. CT was responsible for data acquisition. BS, KC, SK and CT designed the yoga intervention. NS and PW contributed to data acquisition by performing salivary analyses. TT carried out the statistical analyses and wrote the first draft, with support from SK, AH and NC. All authors critically reviewed and revised all drafts and approved and are accountable for the accuracy and integrity of the final version. The corresponding author attests that all listed authors meet authorship criteria and that no others meeting the criteria have been omitted. TT is the guarantor.

**Funding** The study was funded by the UK Medical Research Council (MR/J000175/1). The funders and study sponsors have played no role in the study design; in the collection, analysis and interpretation of data; in the writing of the report; nor in the decision to submit the article for publication. The researchers were independent from the funders and sponsors and all authors had full access to all of the data (including statistical reports and tables) in the study and can take responsibility for the integrity of the data and the accuracy of the data analysis.

**Competing interests** None declared.

**Patient consent for publication** Not required.

**Ethics approval** Ethical approval was granted by Camberwell St Giles Research Ethics Committee (Ref: 12/LO/0597).

**Provenance and peer review** Not commissioned; externally peer reviewed.

**Data availability statement** Data are available upon reasonable request.

**ORCID iD**
Therese Tillin http://orcid.org/0000-0003-1590-9826

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
