## [Reviewer comments · BMJ Open]

ARTICLE DETAILS

TITLE (PROVISIONAL)	Yoga and Cardiovascular Health Trial (YACHT): a UK-based randomised mechanistic study of a yoga intervention plus usual care vs usual care alone following an acute coronary event
AUTHORS	Tillin, Therese; Tuson, Claire; Sowa, Barbara; Chattopadhyay, Kaushik; Sattar, Naveed; Welsh, Paul; Roberts, Ian; Ebrahim, Shah; Kinra, Sanjay; Hughes, A; Chaturvedi, Nishi

VERSION 1 – REVIEW

REVIEWER	Mao Chen Department of Cardiology, West China Hospital of Sichuan University
REVIEW RETURNED	24-Mar-2019

GENERAL COMMENTS	The study investigated the effect of yoga training on subclinical cardiovascular measures in patients with acute coronary syndrome and found no additional benefit from three months of yoga training compared with routine cardiac rehabilitation. The study comprehensively analyzed the effect of yoga training on many cardiovascular measures, but some deficiencies need to be pointed out. 1.The population selected by the author is patients with acute coronary syndrome, among which the proportion of patients with myocardial infarction is not clear. As can be seen from the data in table 1, the proportion of patients with heart failure is only one fifth. Since yoga is a form of cardiac rehabilitation exercise, it may be more valuable to focus the study population on patients with heart failure after myocardial infarction.2.The time interval of coronary events in the population selected by the author was about 50 to 60 days before enrollment. Could the authors indicate the purpose of such consideration? It is interesting to know how many days after the event is safe and effective for patients with acute coronary syndrome to start cardiac rehabilitation training.3.The authors observed the changes in cardiac diastolic function and exercise tolerance of patients in the two groups after 3 months training. We would like to see the change trend and dynamic change of cardiac function and exercise tolerance of patients during the 3-month training.4. The study endpoints should be classified.
---

REVIEWER	Andrew Freeman MD National Jewish Health Denver, CO USA
REVIEW RETURNED	24-Mar-2019

GENERAL COMMENTS	In this small study of 80 patients who were referred to cardiac rehab, were randomized to 3 months of 18-24 yoga classes vs usual cardiac rehab and found no significant differences in post-3-minute step test blood pressures, 6MWT, peak VO₂, and selected autonomic and lipid markers. Some questions:  1) While e/E' is one diastolic parameter, why not use the diastolic dysfunction criteria based on the ASE guidelines? 2) I think it is incorrect to say that the yoga program addition did not change "any" CV parameters. As an example, resting SBP and DBP are often improved in those who practice yoga so I am unclear why this was done after exercise? 3) Other studies (i.e. Mount Abu, Ornish etc) incorporating Yoga techniques often measured mental status, CVD event rate, angina which were neglected in this study 4) Cardiac rehab in the US is usually 36 or up to 72 hours for intensive cardiac rehab – why was this so much less in both groups? 5) Were patients adhering/practicing the yoga techniques outside of the teaching sessions? 6) Marked drop out reduced sample size – why was there such drop out? 7) 6MWT are generally not used for exercise capacity except at a gross level and vary markedly by the test giver. Why not use a symptom limited treadmill stress test? Or do a cardiopulmonary stress test and get VO₂ then? 8) It would be expected yoga may not show an impact on exercise measures – what about inflammation markers? 9) With an N=4 not sure you could make any conclusions 10) 3 months may not be long enough to show CIMT changes, especially in just 25 people As such while this study is interesting, I think the scope and conclusions needs to be reframed using the parameters mentioned above. It would be expected that many of the "physical" parameters checked might not change, but CVD outcomes like angina, NYHA class, CVD event rates etc are very important, as is resting BP and perhaps exercise stress testing.
---

REVIEWER	Raviteja R Guddeti Creighton University School of Medicine, Omaha, USA
REVIEW RETURNED	30-Mar-2019

GENERAL COMMENTS	Tillin T et al reported effects of yoga practice on subclinical cardiovascular measures, risk factors and neuro-endocrine pathways following acute coronary syndrome in the YACHT randomised study. They concluded that a structured 3-month yoga intervention added to usual care cardiac rehabilitation following an acute coronary event provided no additional benefit on any of the cardiovascular outcomes selected for the study. Although the
---

	study adopted a comprehensive approach to measuring cardiovascular outcomes in response to yoga intervention it has several limitations.  1. The study was performed between 2012 and 2014, what was the reason for delay in publishing the results? Are there any long-term data relevant to this study? It is understandable that yoga intervention was performed only for 3 months in this study. 2. Any data on clinical outcomes such as rehospitalization, new onset heart failure etc post-ACS? 3. The study included patients post acute coronary syndrome. What was the treatment for ACS in this study population? How many of the patients received percutaneous coronary intervention and how many were treated medically? 4. What was the severity of coronary artery disease in the study population? How many patients had single-, two- and three-vessel coronary artery disease? 5. As the authors pointed out in the limitations section, this study is severely underpowered due to high drop out rates which will precluded reasonable conclusions to be drawn from the available data. 6. E/e' was chosen as the sole parameter for assessing diastolic function which is not accurate. Diastolic function is assessed using E/A, E/e', e' medial and lateral mitral annulus velocities, tricuspid regurgitation velocity and left atrial volume index. Page 18, Table 2 lines 17 and 18 show E/e' values before and after intervention in both groups. A median value of 9.74 and 8.72 in the yoga + usual care and usual care groups pre-intervention is considered normal (E/e' <14). This points to the fact that these patients did not have any diastolic dysfunction at baseline to benefit from any form of intervention. Similarly, mitral E/A values of 1.02 and 1.16 respectively are also considered normal. And finally left atrial diameter indexed to height is also normal. 7. Table 3, line 21 shows ejection fraction of 54% in both groups which is again normal. It would be interesting to see how intervention benefited those with LV systolic dysfunction. 8. Tables need abbreviations mentioned that the bottom.
--	---

REVIEWER	David de Gonzalo IIBB-CSIC, Barcelona
REVIEW RETURNED	05-Jul-2019

GENERAL COMMENTS	Minor comments: Please indicate which test was used to evaluate normality.
--

REVIEWER	Kai Jin Edinburgh Napier University
REVIEW RETURNED	28-Jul-2019

GENERAL COMMENTS	The statistical analysis is well written including detail information on sample size calculation, clearly defined list of variables and analysis methods. The research question was to examine effects
--

	of yoga measured by any mean/median changes on the cardiovascular outcomes from pre to post differed in the two groups (yoga+usual group VS usual group). These were measured by the time/group interaction in the repeated measures ANOVA and p value for the groups differences for primary outcome results have been provided for interpretation. However, I am not clear for robust regression models: in the last paragraph of the “Statistical methods” (page 8), authors said “robust regression models.... 3-month follow-up values were adjusted for the preintervention value of each Normally distributed measure, to provide adjusted mean (95%CI) values to allow comparison with pre-intervention observations”. Did this mean that robust regression model used to test differences between pre-intervention and 3-month point within the same groups? What about the between group differences? Please include the p value for group differences into the secondary outcome results and table 3 & 4 to help interpretation of results. Please also add p value comparison between before and day of first yoga class for table 5b. Please include the reference for the use robust regression models.
--	---

REVIEWER	Guillien Alicia Team of Environmental Epidemiology applied to Reproduction and Respiratory Health, Inserm, CNRS, University Grenoble Alpes, Institute for Advanced Biosciences (IAB), U1209
REVIEW RETURNED	01-Aug-2019

GENERAL COMMENTS	Comments to the Authors In this article entitled « Yoga and Cardiovascular Health Trial (YACHT): a UK-based randomized controlled trial of a yoga intervention plus usual care vs usual care alone following a coronary event», the authors investigated the benefit of yoga sessions added to usual care on cardiovascular and neuro-endocrine outcomes in patients referred to cardiac rehabilitation. The main results reflect that none of the measured outcomes was significantly improved in the group who performed yoga sessions. The major strength of this study is the parallel-randomized controlled trial design of the study. Moreover, yoga sessions were provided by a teacher certified in yoga and cardiac rehabilitation. However, this article suffers from some weaknesses, both on the form and the substance. Regarding the form, the paper is confusing in many parts and the take home message is unclear. In addition, as a statistical reviewer, I have some concerns on the overall strategy of analysis and statistical methods used.  1. Two primary outcomes are defined and it is not clear which one was used to estimate the necessary sample size. 2. From my point of view, the number of measured outcomes (primary outcomes and secondary) and the number of studies (chronic study and acute study) make the take home message unclear. 3. Regarding the sample size calculation, the authors consider non-adjusted analyses while all there analyses were adjusted on pre-intervention measures. 4. Some of the results are presented as median (95% CI of the median) or median (IQR), which can be confusing. 5. In tables 1, 3, 4 and S1a, p-value should be provided even if they were non significant.
--

	6. Analyses should be adjusted on cofounders. As an example, participation of partner should be taken into account.
--	---

VERSION 1 – AUTHOR RESPONSE

Reviewer: 1

Reviewer Name: Mao Chen

Institution and Country: Department of Cardiology, West China Hospital of Sichuan University Please state any competing interests or state 'None declared': None.

The study investigated the effect of yoga training on subclinical cardiovascular measures in patients with acute coronary syndrome and found no additional benefit from three months of yoga training compared with routine cardiac rehabilitation. The study comprehensively analyzed the effect of yoga training on many cardiovascular measures, but some deficiencies need to be pointed out.

1. The population selected by the author is patients with acute coronary syndrome, among which the proportion of patients with myocardial infarction is not clear. As can be seen from the data in table 1, the proportion of patients with heart failure is only one fifth. Since yoga is a form of cardiac rehabilitation exercise, it may be more valuable to focus the study population on patients with heart failure after myocardial infarction.

The study was designed to examine mechanisms underlying any beneficial effects of yoga over the usual rehabilitation programmes provided in people who are eligible for cardiac rehabilitation in UK. Cardiac rehabilitation programmes in the UK are offered to patients with acute MI, revascularisation procedures, CABG, valve surgery and heart failure, and are not limited to those with heart failure. However, our study protocol specified referral to cardiac rehabilitation following angioplasty, CABG or prescribed medical management as a treatment for an acute coronary syndrome -and not heart failure per se (we have clarified this in the manuscript title. Please also see methods: study population page 4)

2. The time interval of coronary events in the population selected by the author was about 50 to 60 days before enrollment. Could the authors indicate the purpose of such consideration? It is interesting to know how many days after the event is safe and effective for patients with acute coronary syndrome to start cardiac rehabilitation training.

This was a pragmatic decision. The research team had to visit several sites in order to identify suitable participants who had already been referred to cardiac rehabilitation programs, then to make contact with them and gain informed consent. The decision to refer a person for cardiac rehabilitation was taken by the clinical care teams who were independent of this study and followed local programs (please see supplementary material YACHT study package, pages 23-29) based on national guidelines. Current UK guidelines (NICE, the Department of Health and the BACPR) encourage starting CR in 28 – 42 days (latter for CABG); however in practice few health authorities achieve this goal – the average delay to initiation of cardiac rehabilitation following MI in London was approximately 50 days in 2012-3 (1) which is fairly representative of the UK at that time. Hence our time interval to recruitment represented the minimum practicable interval before recruitment.

3. The authors observed the changes in cardiac diastolic function and exercise tolerance of patients in the two groups after 3 months training. We would like to see the change trend and dynamic change of cardiac function and exercise tolerance of patients during the 3-month training.

Unfortunately it is not possible to provide these data. The study was designed to assess outcomes before and at the end of the 3 month yoga vs usual cardiac rehabilitation programs. By design, no interim assessments were performed during the 3 month period with the exception of the acute study which took place in the yoga group only on the day of the first yoga training session and measured blood pressure, heart rate and salivary amylase and cortisol pre and post exercise testing, plus estimates of peak oxygen consumption post-exercise, both before and after the first yoga session.

4. The study endpoints should be classified.

We have listed primary and secondary outcomes for the chronic study (before and after 3 months of intervention) and for the acute study (on the day of first yoga training) (pages 5-7)

Reviewer: 2

Reviewer Name: Andrew Freeman MD

Institution and Country: National Jewish Health Denver, CO USA Please state any competing interests or state 'None declared': None

In this small study of 80 patients who were referred to cardiac rehab, were randomized to 3 months of 18-24 yoga classes vs usual cardiac rehab and found no significant differences in post-3-minute step test blood pressures, 6MWT, peak VO₂, and selected autonomic and lipid markers.

Some questions:

1) While e/E' is one diastolic parameter, why not use the diastolic dysfunction criteria based on the ASE guidelines?

At the time the study was designed the ASE recommendations (2) for estimation of filling pressure in individuals with preserved ejection fraction were that 'the E/e' ratio should be calculated' (page 127). In addition this measure was chosen as E/e' had been demonstrated to predict survival after myocardial infarction as we now make clear on page 3. We recognize that subsequent guidelines have been modified and have extended these recommendations but these recommendations came after the finalization of the statistical analysis plan. Nevertheless, how best to diagnose diastolic dysfunction is still debated and current methods are unreliable (see for example (3, 4)). Other measures of diastolic function are reported as secondary outcomes in Table 3 and described on page 6, including mitral E:A ratio, e' peak velocity and left atrial diameter indexed to height. We have now added these to the text on page 10 describing results for secondary outcomes

2) I think it is incorrect to say that the yoga program addition did not change "any" CV parameters. As an example, resting SBP and DBP are often improved in those who practice yoga so I am unclear why this was done after exercise?

We apologize if this was unclear and we have reworded the revised manuscript to make it clearer (chronic study, secondary outcomes para 2, page 5, Table 3, page 20). Assessment of blood pressure was performed at rest in all participants and following exercise before and after the initial yoga session (Table 5a – acute study). Exercise blood pressure was assessed in addition to resting blood pressure since abnormal BP responses to exercise predict risk of future cardiovascular events and mortality independent of resting BP and other cardiovascular risk factors. (5) Based on our findings there was no evidence that blood pressure (resting or exercise) was reduced by yoga in the acute setting. We also measured resting seated brachial blood pressure and 24 hour ambulatory blood pressure before and after the 3 months of yoga intervention - these data are reported in table 3. – while blood pressure fell in both yoga and usual care groups there was no evidence of a reduction in blood pressure attributable to yoga.

3) Other studies (i.e. Mount Abu, Ornish etc) incorporating Yoga techniques often measured mental status, CVD event rate, angina which were neglected in this study

The study was designed to provide mechanistic insights into any beneficial effects of yoga and was not intended or powered to examine CVD outcomes. We did examine responses to the 10 item perceived stress self-completion questionnaire and included a global rating of health status (EQ5D) which includes an anxiety and depression domain, before and after the three month intervention - these are reported in table 3. There was a small improvement in the yoga+usual care group for the EQ5D health status score with no change for the usual care group on this measure, although confidence intervals were wide in both groups. The EQ5D self-rated health 'thermometer' improved slightly but to equal extents in both treatment groups and the perceived stress score was little changed in either group (Table 3 and description on page 10 under 'other measures'). We have revised the manuscript to clarify these points (page 10).

4) Cardiac rehab in the US is usually 36 or up to 72 hours for intensive cardiac rehab – why was this so much less in both groups?

This is a UK study where usual cardiac rehabilitation was administered by National Health Service providers following national guidelines from the UK National Institute for Health and Care Excellence (NICE), see appended material (Yacht study package, pages 23-29) - the rehabilitation programme described was routine care provided alongside, but separately from, this study.

5) Were patients adhering/practicing the yoga techniques outside of the teaching sessions?

A 'prescription' of yoga practice with an accompanying DVD was provided for practice at home (page 4) - however, we are unable to verify whether participants practiced yoga outside the designated sessions and cannot comment further on this point.

6) Marked drop out reduced sample size – why was there such drop out?

Reasons are noted in the Consort flow diagram under 'follow-up patients' and described under results on page 9, also in tables S1a/ S1b. We acknowledge that drop-out is greater in the yoga +usual care arm, possibly reflecting the dual burden of attending both yoga and usual care rehabilitation

7) 6MWT are generally not used for exercise capacity except at a gross level and vary markedly by the test giver. Why not use a symptom limited treadmill stress test? Or do a cardiopulmonary stress test and get VO2 then?

Clinician advice was to avoid repeated treadmill/stress testing for safety reasons. References 17 and 18 describe validation of the 6 minute walk test in outpatient cardiac rehabilitation

8) It would be expected yoga may not show an impact on exercise measures – what about inflammation markers?

Unfortunately inflammation markers were not measured, although salivary cortisol which could be considered a marker of stress did not show evidence of yoga related benefits (Table 3)

9) With an N=4 not sure you could make any conclusions

We are not sure if you mean the addition of 4 people who did not complete the requisite 24 yoga classes to the yoga arm results (page 10)? - this was performed only as a sensitivity analysis to check

whether addition of their findings would alter our main results for the yoga plus usual care arm. Alternatively you may be referring to the inclusion of the same 4 people in supplemental table S1b which lists characteristics of those who did and did not complete the three month follow-up period- in both cases we have tried not to draw conclusions but to show the available information

10) 3 months may not be long enough to show CIMT changes, especially in just 25 people

We agree with this point and have added a comment under 'strengths and limitations' in the discussion.

As such while this study is interesting, I think the scope and conclusions needs to be reframed using the parameters mentioned above. It would be expected that many of the "physical" parameters checked might not change, but CVD outcomes like angina, NYHA class, CVD event rates etc are very important, as is resting BP and perhaps exercise stress testing.

We accept this point and have revised the MS to make this clear. We have clarified the point regarding resting BP above. We have clarified that the study was not intended to or powered to measure CVD outcomes and exercise stress testing was not considered appropriate in the setting of this research design.

Reviewer: 3

Reviewer Name: Raviteja R Guddeti

Institution and Country: Creighton University School of Medicine, Omaha, USA Please state any competing interests or state 'None declared': None declared

Tillin T et al reported effects of yoga practice on subclinical cardiovascular measures, risk factors and neuro-endocrine pathways following acute coronary syndrome in the YACHT randomised study. They concluded that a structured 3-month yoga intervention added to usual care cardiac rehabilitation following an acute coronary event provided no additional benefit on any of the cardiovascular outcomes selected for the study. Although the study adopted a comprehensive approach to measuring cardiovascular outcomes in response to yoga intervention it has several limitations.

1. The study was performed between 2012 and 2014, what was the reason for delay in publishing the results? Are there any long-term data relevant to this study? It is understandable that yoga intervention was performed only for 3 months in this study.

The study was designed as a mechanistic parallel study to the larger Indian Council for Medical Research and Medical Research Council, UK funded study of yoga as a primary method of cardiac rehabilitation in India, and we delayed publication until closer to the time of the main study results. As noted above, the study did not include long-term follow up in its aims and indeed was not powered to do so.

2. Any data on clinical outcomes such as rehospitalization, new onset heart failure etc post-ACS?

These data are not available and the study was not designed or sufficiently powered to examine these longer term outcomes

3. The study included patients post acute coronary syndrome. What was the treatment for ACS in this study population? How many of the patients received percutaneous coronary intervention and how many were treated medically?

The majority (73 out of 80) had received PCI which is consistent with current practice in UK. We have added this information to the revised manuscript.

4. What was the severity of coronary artery disease in the study population? How many patients had single-, two- and three-vessel coronary artery disease?

Unfortunately we do not have access to this information.

5. As the authors pointed out in the limitations section, this study is severely underpowered due to high drop out rates which will precluded reasonable conclusions to be drawn from the available data.

In the revised manuscript we acknowledge the high levels of drop-out which limit our power to detect small benefits of yoga. However, there is little indication across the primary and multiple secondary endpoints of additional benefits of yoga plus usual care over usual care alone. We now add estimates of the between group differences in each primary and secondary outcome measure at follow-up with 95% CIs and suggest that the effect sizes for each measure are unlikely to exceed the 95% CIs for each difference.

6. E/e' was chosen as the sole parameter for assessing diastolic function which is not accurate. Diastolic function is assessed using E/A, E/e', e' medial and lateral mitral annulus velocities, tricuspid regurgitation velocity and left atrial volume index. Page 18, Table 2 lines 17 and 18 show E/e' values before and after intervention in both groups. A median value of 9.74 and 8.72 in the yoga + usual care and usual care groups pre-intervention is considered normal (E/e' <14). This points to the fact that these patients did not have any diastolic dysfunction at baseline to benefit from any form of intervention. Similarly, mitral E/A values of 1.02 and 1.16 respectively are also considered normal. And finally left atrial diameter indexed to height is also normal.

We agree that diastolic function is a complex multifaceted process and not easily captured by any single measure, although for trial purposes a single endpoint needs to be defined. In the revised manuscript we have clarified that E/e' was chosen as a simple estimate of left ventricular filling pressure, which is an index of one aspect of diastolic function. We also clarify our reasons for choosing this measure (page 3). Other measures relevant to diastolic function, E/A, e' (medial and lateral wall average) and left atrial diameter index are included in the manuscript and reported under secondary outcomes (table 3, pages 6 and 10). How best to diagnose diastolic dysfunction is debated and current methods are unreliable (see for example (3, 4)). We respectfully disagree with the reviewer that a value of E/e' <14 is normal. Impaired function is a continuum not a binary condition. Based on NORRE data it is true that E/e' is rarely >14 in healthy individuals but this does not imply the converse – namely that E/e' ≤14 is healthy. The average value for E/e' in healthy individuals from the relevant age group (4-60) in NORRE was 6.8+1.8, and the original data from Nagueh et al.(6) shows a clear linear relationship between E/e' and invasively measured filling pressure across the entire range from 5 to 30. More recent multicenter data (7) are consistent with this. It is also relevant that increased E/e' predicts increased risk of cardiovascular risk across the entire range of E/e' values (8, 9) (not only for people with E/e' >14 or 15) and that intermediate values of E/e' (i.e. between 8 and 14) are associated with significantly elevated risk in the general population (intermediate between those with E/e' <8 and those with E/e' >14) .(9) Approximately half of participants in this study fell within this intermediate category of E/e'. We cannot agree with the reviewer that only people with unequivocally abnormal diastolic function (i.e. E/e' >14) would be expected to show any benefit from an intervention even in terms of E/E' since there is evidence that exercise improves diastolic function even in healthy individuals.(10)

7. Table 3, line 21 shows ejection fraction of 54% in both groups which is again normal. It would be interesting to see how intervention benefited those with LV systolic dysfunction.

An ejection fraction in the range 45-54% is classified as 'mildly abnormal' according to ASE guidelines.(11) Consistent with this a recent multicenter study reported that the lower limit of the normal range for Europeans is 55.8% in men and 57.3% in women.(12) On average participants in this study had preserved ejection fractions but it would be incorrect to consider them normal. Pre-intervention, there were 16 people in the usual care arm and 8 in the yoga+plus usual care arm with EF<50%. Given the relatively small number of people with reduced or mildly reduced ejection fraction we do not believe that an unplanned subgroup analysis would be appropriate .

8. Tables need abbreviations mentioned that the bottom.

These have been added

Reviewer: 4

Reviewer Name: David de Gonzalo

Institution and Country: IIBB-CSIC, Barcelona Please state any competing interests or state 'None declared': None declare

Minor comments: Please indicate which test was used to evaluate normality.

A specific test for normality was not used, but histograms and Normal Q-Q plots were examined for evidence of departure from normality

Reviewer: 5

Reviewer Name: Kai Jin

Institution and Country: Edinburgh Napier University Please state any competing interests or state 'None declared': None declared

The statistical analysis is well written including detail information on sample size calculation, clearly defined list of variables and analysis methods. The research question was to examine effects of yoga measured by any mean/median changes on the cardiovascular outcomes from pre to post differed in the two groups (yoga+usual group VS usual group). These were measured by the time/group interaction in the repeated measures ANOVA and p value for the groups differences for primary outcome results have been provided for interpretation.

However, I am not clear for robust regression models: in the last paragraph of the "Statistical methods" (page 8), authors said "robust regression models.... 3-month follow-up values were adjusted for the preintervention value of each Normally distributed measure, to provide adjusted mean (95%CI) values to allow comparison with pre-intervention observations". Did this mean that robust regression model used to test differences between pre-intervention and 3-month point within the same groups? What about the between group differences? Please include the p value for group differences into the secondary outcome results and table 3 & 4 to help interpretation of results. Please also add p value comparison between before and day of first yoga class for table 5b.

We apologise for any confusion. The comparison was only between the two groups at 3 months following intervention. However, the adjusted mean values were provided to aid interpretation when viewing the table. We have clarified this in the revised text.

We had elected not to show p values for secondary outcomes due to the large number of statistical tests that would be involved and in line with some current thinking on the use of p values. 95% CIs

are presented in all results tables. However, if the editors are in agreement then we will be pleased to provide p values for secondary outcomes. Please note that we have now added the between group differences(95% CI) at follow-up to each table, which we hope will aid interpretation .

Please include the reference for the use robust regression models.

We have now added this: Huber PJ, Ronchetti E. Robust statistics. 2nd ed. ed. Oxford: Wiley-Blackwell; 2009

Reviewer: 6

Reviewer Name: Guillien Alicia

Institution and Country: Team of Environmental Epidemiology applied to Reproduction and Respiratory Health, Inserm, CNRS, University Grenoble Alpes, Institute for Advanced Biosciences (IAB), U1209 Please state any competing interests or state 'None declared': None declared

In this article entitled « Yoga and Cardiovascular Health Trial (YACHT): a UK-based randomized controlled trial of a yoga intervention plus usual care vs usual care alone following a coronary event», the authors investigated the benefit of yoga sessions added to usual care on cardiovascular and neuro-endocrine outcomes in patients referred to cardiac rehabilitation. The main results reflect that none of the measured outcomes was significantly improved in the group who performed yoga sessions. The major strength of this study is the parallel-randomized controlled trial design of the study. Moreover, yoga sessions were provided by a teacher certified in yoga and cardiac rehabilitation. However, this article suffers from some weaknesses, both on the form and the substance. Regarding the form, the paper is confusing in many parts and the take home message is unclear. In addition, as a statistical reviewer, I have some concerns on the overall strategy of analysis and statistical methods used.

1. Two primary outcomes are defined and it is not clear which one was used to estimate the necessary sample size.

In the design phase we had planned to use the larger estimate of the two co-primary endpoints for the sample size but as it transpired the minimally clinically important differences for both co-primary outcomes were similar (0.5 SD) so the estimate applies to both E/e' ratio and for the distance walked in the 6 minute walk test. We have reworded the sample size section to attempt to clarify this.

2. From my point of view, the number of measured outcomes (primary outcomes and secondary) and the number of studies (chronic study and acute study) make the take home message unclear.

We have revised the manuscript, particularly the abstract and conclusion to make the message clearer.

3. Regarding the sample size calculation, the authors consider non-adjusted analyses while all there analyses were adjusted on pre-intervention measures.

We have revised the wording to clarify that to improve the precision of treatment effect estimates and reduce the sample size requirements, the sample size estimates were based on the analyses adjusted on pre-intervention measures.

4. Some of the results are presented as median (95% CI of the median) or median (IQR), which can be confusing.

Apologies, we have now corrected table 1 to show median(95%CI)

5. In tables 1, 3, 4 and S1a, p-value should be provided even if they were non significant.

As noted above we limited use of p values to between group differences in primary outcomes, given the number of secondary outcomes and some current thinking on the use of p values. However, we are happy to add p values to tables 1, 3,4 and 5 if the editors are in agreement. We have not added p values to table S1A where there number of drop-outs from the usual care group is only 5 as we think it inappropriate to provide p values for such small numbers (these data are only shown for completeness). We have also added between group differences (95% CI) at follow-up to all outcome measures, which we hope will aid interpretation

6. Analyses should be adjusted on cofounders. As an example, participation of partner should be taken into account.

Due to randomisation, the effects of the interventions are not confounded. We feel that it would be inappropriate to adjust for an intermediate variable (e.g. partner participation) which is on the causal pathway from the intervention to the outcome as this would introduce collider bias (i.e. overadjustment). However, we agree that it may be helpful to adjust for informative baseline covariates (i.e. auxillary variables) and have performed additional sensitivity analyses adjusting primary outcomes as follows: E'le and step test: age, sex, diabetes, hypertension, body mass index. 6 minute walk test: age, sex, diabetes, body mass index and height. Findings are summarized in the text under 'sensitivity analyses, page 13)

VERSION 2 – REVIEW

REVIEWER	Mao Chen West China Hospital, Sichuan University
REVIEW RETURNED	16-Sep-2019

GENERAL COMMENTS	As I have stated before, I think that the study did not include a group of patients high-risk enough who could gain greatest benefit from yoga intervention. As the study period was such short and the dropout rate was so high, the choice of general ACS patients according to cardiac rehabilitation guidelines was not optimal for intervention trial. To resolve the problem, a two-by-two factorial design trial could be performed. The ACS patients could be randomly assign to receive usual care alone or non-cardiac rehabilitation and to receive yoga or non-cardiac rehabilitation. It is conducive to clarify the independent and superimposed effect of short-term yoga intervention on cardiovascular health.
---

REVIEWER	Andrew Freeman National Jewish Health
REVIEW RETURNED	03-Sep-2019

GENERAL COMMENTS	Thanks for your clarifications: 1) Your conclusion is again incorrect. "We found no evidence that a structured 3-month yoga intervention added to usual care following an acute coronary event improved any cardiovascular or neuro-endocrine measures."
--

	Should be something like: In a small pilot study of 60 participants, with only 25 in the intervention group, in the small subset of measures taken, there was no discernable improvement in diastology, 6MWT, etc 2) Still no discussion of other studies Ornish, Mount Abu, others showing benefits. I think this could be publishable in a pilot form but the conclusions need to be modified.
--	---

REVIEWER	Kai Jin Edinburgh Napier University
REVIEW RETURNED	19-Sep-2019

GENERAL COMMENTS	The reviewer completed the checklist but made no further comments.
--

VERSION 2 – AUTHOR RESPONSE

Reviewer: 2

Reviewer Name: Andrew Freeman

Institution and Country: National Jewish Health Please state any competing interests or state 'None declared': None

Please leave your comments for the authors below Thanks for your clarifications:

1) Your conclusion is again incorrect.

"We found no evidence that a structured 3-month yoga intervention added to usual care following an acute coronary event improved any cardiovascular or neuro-endocrine measures."

Should be something like:

In a small pilot study of 60 participants, with only 25 in the intervention group, in the small subset of measures taken, there was no discernable improvement in diastology, 6MWT, etc

>>We have modified the conclusion in both abstract and main text as follows:

'In this small UK-based randomised mechanistic study, with 60 completing participants (of whom 25 were in the yoga+usual care group), we found no discernible improvement associated with the addition of a structured 3 month yoga intervention to usual cardiac rehabilitation care in key cardiovascular and neuroendocrine measures shown to be responsive to yoga in previous mechanistic studies'

We hope that this is an acceptably cautious conclusion. The study was not designed as a pilot study and we felt that given the absence of discernible benefits across the very wide range of primary and secondary outcome measures, it would not be appropriate to describe this as 'a small subset of measures'.

As stated under 'strengths and limitations', our study was designed as a parallel mechanistic study to complement a larger (around 4000 patients) Indian Council for Medical Research and Medical Research Council, UK funded study of yoga as a primary method of CR in India.

2) Still no discussion of other studies Ornish, Mount Abu, others showing benefits.

>>We have added both to the discussion (refs 55, 56, 59, 60) We had also already included in the discussion a number of studies with positive yoga-related outcomes (e.g. references: 8, 10, 39, 44, 50-53, 58)

I think this could be publishable in a pilot form but the conclusions need to be modified.

Reviewer: 1

Reviewer Name: Mao Chen

Institution and Country: West China Hospital, Sichuan University Please state any competing interests or state 'None declared': None declared.

Please leave your comments for the authors below: As I have stated before, I think that the study did not include a group of patients high-risk enough who could gain greatest benefit from yoga intervention. As the study period was such short and the dropout rate was so high, the choice of general ACS patients according to cardiac rehabilitation guidelines was not optimal for intervention trial. To resolve the problem, a two-by-two factorial design trial could be performed. The ACS patients could be randomly assign to receive usual care alone or non-cardiac rehabilitation and to receive yoga or non-cardiac rehabilitation. It is conducive to clarify the independent and superimposed effect of short-term yoga intervention on cardiovascular health.

>>In the UK, according to evidence-based national guidelines, all patients who have experienced an acute coronary event, revascularization procedure or coronary artery bypass grafting are offered as standard care 6-12 weeks of cardiac rehabilitation ('usual care') and hence the 3 month time period for the study intervention. As noted under 'strengths and limitations' it would not be ethical to offer yoga alone or non-cardiac rehabilitation alone, hence we cannot conduct the comparisons suggested by the reviewer.

The research question was to ascertain whether yoga would be associated with additional improvements in cardiovascular function and exercise capacity both chronically and acutely in people eligible for usual cardiac rehabilitation. Lack of evidence meant that we had no prior beliefs which would have led us to restrict the yoga +usual care intervention to selected high risk individuals and indeed this was not within the remit of the study which sought to determine whether yoga could be a useful adjunct to usual cardiac rehabilitation. The pre-specified inclusion criteria were as stated under Methods, 'Study population'

The drop-out rate is acknowledged in the results and under 'Strengths and Limitations. While it is true that the drop-out rate from the yoga+usual care groups was higher than from the usual care alone group, we did have 35 usual care completers and 25 yoga + usual care completers (compared with a sample size estimate of 33 in each group to enable detection of 0.5SD between group difference in primary outcomes). This is noted under 'strengths and limitations', along with comment that 'although given the measured effect sizes and confidence intervals, we believe if there are benefits of yoga on the measured outcomes, they are likely to be small'.

Reviewer: 5

Reviewer Name: Kai Jin

Institution and Country: Edinburgh Napier University Please state any competing interests or state 'None declared': None declared

Please leave your comments for the authors below I have no further comments.

VERSION 3 – REVIEW

REVIEWER	Mao Chen West China Hospital of Sichuan University, China
REVIEW RETURNED	13-Oct-2019
GENERAL COMMENTS	Although the study has many shortcomings in study design, inclusion criteria, and performance, the authors have described the results objectively and pointed out those in study limitations.